# LEARNING TO PROMPT SEGMENTATION FOUNDATION MODELS

## ABSTRACT

Segmentation foundation models (SFMs) like SEEM and SAM have demonstrated great potential in learning to segment anything. The core design of SFMs lies with "Promptable Segmentation", which takes a handcrafted prompt as input and returns the expected segmentation mask. SFMs work with two types of prompts including spatial prompts (e.g., points) and semantic prompts (e.g., texts), which work together to prompt SFMs to segment anything on downstream datasets. Despite the important role of prompts, how to acquire suitable prompts for SFMs is largely under-explored. In this work, we examine the architecture of SFMs and identify two challenges for learning effective prompts for SFMs. To this end, we propose spatial-semantic prompt learning (SSPrompt) that learns effective semantic and spatial prompts for better SFMs. Specifically, SSPrompt introduces spatial prompt learning and semantic prompt learning, which optimize spatial prompts and semantic prompts directly over the embedding space and selectively leverage the knowledge encoded in pre-trained prompt encoders. Extensive experiments show that SSPrompt achieves superior image segmentation performance consistently across multiple widely adopted datasets.

## 1 INTRODUCTION

Recently, Segmentation Foundation Models (SFMs), such as Segment Everything Everywhere Model (SEEM) (Zou et al., 2023) and Segment Anything Model (SAM) (Kirillov et al., 2023), have achieved striking image segmentation performance over various downstream datasets (Cordts et al., 2016; Zhou et al., 2017), demonstrating their great potential in learning to segment anything. The core design lies with "Promptable Segmentation", i.e., SFMs take handcrafted prompts as inputs and return expected segmentation masks. Generally, SFMs work with two types of prompts including spatial prompts (e.g., points or bounding boxes represented by 2D coordinates) and semantic prompts (e.g., free-form texts represented by word tokens), which work together to prompt SFMs to identify and segment anything in images. However, directly using default prompts (i.e., raw class names as the semantic prompts and a grid of points as the spatial prompts) for every downstream dataset is usually sup-optimal, and how to acquire suitable prompts for SFMs is a non-trivial task as a slight modification of prompts could lead to very different segmentation outcome.

By examining the architecture of SFMs in Figure 1, we identify two challenges of learning effective prompts for SFMs: (1) *Limited Search Space in Spatial Prompt Learning.* SFMs take XY coordinates in images as spatial prompts, but optimizing such spatial prompts in low-dimensional space (i.e., two dimensions in XY coordinate system) suffers from the limited search space (Köppen, 2000; Zimek et al., 2012) which could lead to sub-optimal spatial prompts. (2) *Side Effects from Text Prompt Encoder.* Text prompt encoders in SFMs (e.g., CLIP in SAM and UniCL/Florence in SEEM) are largely pre-trained with object-centric image-text data, where the text data is dominated by the description of foreground objects, leading to well-learnt foreground text knowledge but relatively poorly-learnt background text knowledge. Consequently, learning semantic prompts with such text prompt encoders can benefit from the well-learnt text knowledge, but may also suffer from the side effects from the poorly-learnt text knowledge.

In this work, we strive for an effective prompt learning technique for SFMs by addressing the above two issues, aiming to acquire optimal spatial and semantic prompts for downstream segmentation datasets with few-shot data. Considering the architecture of SFMs shown in Figure 1, we argue that

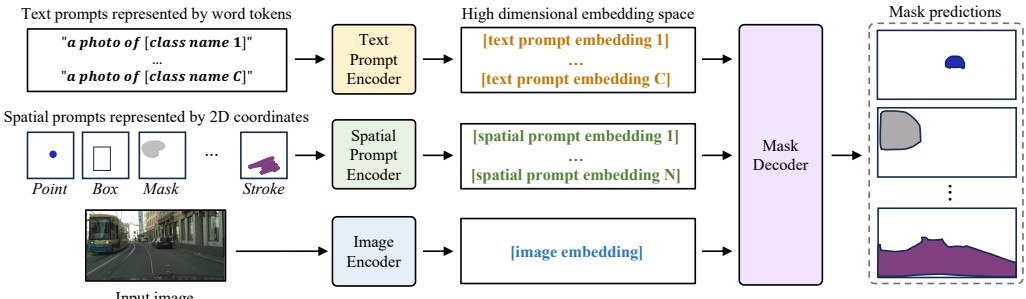

Figure 1: The architecture of segmentation foundation models (SFMs). SFMs (Kirillov et al., 2023; Zou et al., 2023) consist of three core parts: (1) a large *Image Encoder* that encodes input images into image embeddings; (2) prompt encoders including a large *Text Prompt Encoder* that encodes text tokens into text prompt embeddings and a lightweight *Spatial Prompt Encoder* that encodes 2D spatial coordinates into spatial prompt embeddings; and (3) a lightweight *Spatial Prompt Encoder* that predicts the expected segmentation masks based on the image and prompt embeddings.

one effective manner to learn prompts for SFMs is by optimizing prompts directly on the embedding space[1]. Intuitively, optimizing spatial prompts directly on the embedding space could relax limited search space, because embedding space is high-dimensional (e.g., 512D) and has much larger search space as compared with 2-dimensional XY coordinate space. Regarding the Side Effects from Text Prompt Encoder, we argue that the knowledge in text encoder should be utilized selectively so as to benefit from its well-learnt knowledge and concurrently mitigate potential negative effects from its poorly-learnt knowledge.

To this end, we design spatial-semantic prompt learning (SSPrompt) that introduces spatial prompt learning (SpaPrompt) and semantic prompt learning (SemPrompt) for learning effective prompts for SFMs, as illustrated in Figure 2. For semantic prompt learning, SemPrompt employs learnable weights to weight the default semantic prompt embeddings (encoded by fixed *Text Prompt Encoder*) and then fuses the weighted embeddings with a set of *Learnable Semantic Prompt Embeddings* to acquire new semantic prompts. Intuitively, SemPrompt 1) is efficient as its optimization only involves the embeddings encoded by the large text prompt encoder instead of the text prompt encoder itself, and 2) can mitigate potential side effects from the text prompt encoder by introducing learnable weights to selectively leverage the knowledge encoded in the encoder (i.e., the default semantic prompt embeddings encoded by the encoder). For spatial prompt learning, SpaPrompt employs learnable weights to weight the default spatial prompt embeddings (encoded by the fixed spatial prompt encoder) and fuses the weighted embeddings with a set of learnable spatial prompt embeddings to acquire new spatial prompts. In this way, SpaPrompt relaxes the limited search space by optimizing spatial prompts on high-dimensional embedding space. Similar to SemPrompt, SpaPrompt can selectively utilize the knowledge encoded in spatial prompt encoder.

The contributions of this work can be summarized in three major aspects. First, we identify two challenges in prompt learning in SFMs and investigate how to tackle them for the first time to the best of our knowledge. Second, we design spatial-semantic prompt learning which directly optimizes spatial and semantic prompts in the embedding space and selectively exploit the knowledge encoded in prompt encoders, ultimately learning effective prompts for SFM using few-shot data only. Third, extensive experiments show that the proposed method achieves state-of-the-art performances consistently over multiple widely adopted segmentation datasets.

## 2 RELATED WORK

**Segmentation Foundation Models** (SFMs) have recently demonstrated great potential in learning to segment anything (Kirillov et al., 2023; Zou et al., 2023), which achieve striking image segmentation performance over various downstream datasets. To our knowledge, the recent breakthroughs of

---

[1]Following SAM (Kirillov et al., 2023) and SEEM (Zou et al., 2023), in this paper, "embedding" refers to the representation after the encoder. And "text tokens" refer to the text representation before the text encoder.

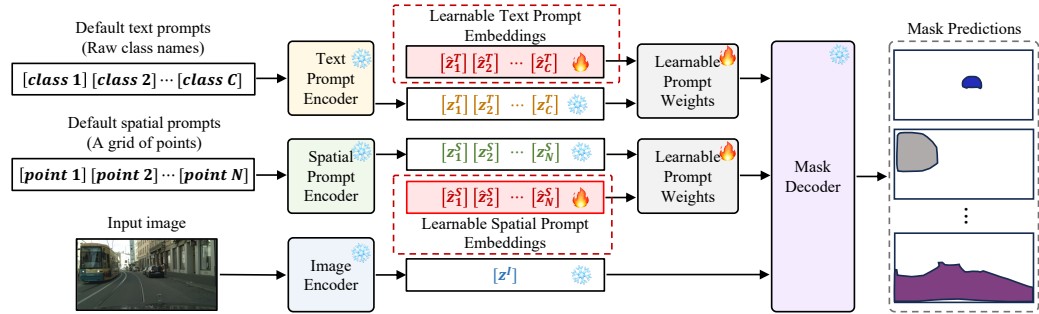

Figure 2: **The framework of semantic-spatial prompt learning (SSPrompt).** SSPrompt optimizes spatial and semantic prompts directly on the embedding space and selectively leverages the knowledge encoded in prompt encoders: it employs learnable weights to weight the default prompt embeddings ($\{z_n^S\}_{n=1}^N$ and $\{z_c^T\}_{c=1}^C$) and fuses the weighted embeddings with the learnable prompt embeddings (i.e., $\{\hat{z}_n^S\}_{n=1}^N$ and $\{\hat{z}_c^T\}_{c=1}^C$) to acquire new prompts. During training, only the *Learnable Prompt Embeddings* and the *Learnable Prompt Embeddings* are updated (marked by Flame), while all rest are frozen (marked by Snowflake).

SFMs, particularly SAM (Kirillov et al., 2023) and SEEM (Zou et al., 2023), are largely driven by the advanced design called "Promptable Segmentation", i.e., SFMs take a handcrafted prompt as input and return the expected segmentation mask. Generally, SFMs involve two types of prompts including semantic prompts (e.g., free-form texts) and spatial prompts (e.g., points or bounding boxes), which provide semantic and spatial information respectively and together prompt segmentation models to identify and segment anything in images (Kirillov et al., 2023; Zou et al., 2023). On the other hand, how to acquire suitable prompts for SFMs is a non-trivial task but largely under-explored. In this work, we focus on investigating how to learn effective prompts for SFMs using few-shot data, aiming to facilitate the deployment of SFMs for task-specific or domain-specific downstream datasets.

**Prompt Learning** aims to adapt foundation models towards downstream tasks by optimizing prompts using few-shot data. In recent years, prompt learning has been extensively studied for NLP foundation models and image classification foundation models (CFMs). Specifically, for NLP foundation models that generally work in a question answering manner, various prompt learning methods have been introduced to learn effective context text tokens to append and improve the raw questions, such as text mining/paraphrasing (Jiang et al., 2020), gradient-based searching (Shin et al., 2020), continuous text token optimization (Zhong et al., 2021; Li & Liang, 2021; Lester et al., 2021; Liu et al., 2023a). On the other hand, for CFMs that classify images based on class names, a variety of prompt learning methods (Zhou et al., 2022b;a; Parisot et al., 2023) have been proposed to learn effective context text tokens to append and improve the raw class names, such as continuous text token optimization (Zhou et al., 2022b), conditional text token optimization (Parisot et al., 2023), etc. Different from previous works that focus on image classification foundation models and optimize text tokens to prompt text encoder, we examine the architecture of segmentation foundation models (SFMs) and propose a more efficient and effective prompt learning method for SFMs. The method directly optimizes spatial and semantic prompts in the embedding space and selectively exploits the knowledge encoded in prompt encoder [2], ultimately learning effective prompt embeddings to prompt the mask decoder.

## 3 METHOD

In this section, we first introduce the background of segmentation foundation models (Section 3.1) and revisit the prompt learning methods of image classification foundation models (Section 3.2). Then, we elaborate our proposed prompt learning method for segmentation foundation models (Section 3.3).

---

[2]Following SAM (Kirillov et al., 2023) and SEEM (Zou et al., 2023), in this paper, "embedding" refers to the representation after the encoder. And "text tokens" refer to the text representation before the text encoder.

### 3.1 Preliminaries of Segmentation Foundation Models

Segmentation Foundation Models (SFMs) (Zou et al., 2023; Kirillov et al., 2023) learn to segment anything by introducing a new "Promptable Segmentation" scheme, where the segmentation model predicts the expected segmentation mask for a given prompt. In this way, SFMs could segment anything (e.g., any objects and background stuff) given proper prompts. Besides, SFMs enable "interactive segmentation" that helps scale up the segmentation training data by using a data engine with model-in-the-loop annotating, which in turn facilitates training more powerful SFMs for better "interactive segmentation".

Specifically, SFMs (Zou et al., 2023; Kirillov et al., 2023) consist of three core parts: (1) an image encoder that encodes images into image embeddings; (2) a text prompt encoder and a spatial prompt encoder, which respectively encode text prompts and spatial prompts into prompt embeddings; (3) a mask decoder that returns the expected segmentation mask based on the image and prompt embeddings.

Given an input image $x^I \in \mathbb{R}^{H \times W \times 3}$ and a set of prompts (e.g., a spatial prompt $x^S$ and a text prompt $x^T$), SFMs first employ the image encoder $Encoder^I$, the spatial prompt encoder $Encoder^S$ and the text prompt encoder $Encoder^T$ to encode them into $D$-dimensional embeddings: $z^I = Encoder^I(x^I)$, $z^S = Encoder^S(x^S)$ and $z^T = Encoder^T(x^T)$, respectively. Then, given the encoded image and prompts, the mask decoder of SFMs predicts the expected segmentation mask:

$$(m, c) = Decoder(z^I | z^S, z^T), \tag{1}$$

where $m$ stands for a predicted binary segmentation mask and $c$ denotes the predicted confidence score of $m$. In SEEM (Zou et al., 2023), $c$ stands for the probability of mask $m$ belonging to the category denoted by text $x^T$. In SAM (Kirillov et al., 2023), when prompted by text prompts, $c$ can also denote the probability of mask $m$ belonging to the category denoted by text $x^T$. When prompted by spatial prompts, $c$ in SAM is class-agnostic and only denotes the quality of predicted mask $m$.

Note, for spatial prompt $x^S$, we mainly consider the format of point, i.e., $x^S = (h, w)$ ($h \in (0, H)$ and $w \in (0, W)$ where $H$ and $W$ denote image height and image width respectively), because all other formats of spatial prompts can be represented in terms of points, e.g., the bounding box can be denoted by two corner points and the coarse mask can be denoted by a set of points.

**Zero-shot Cross-Dataset Inference.** Given an image $x^I$ and a set of default prompts (i.e., raw class names $X_{\text{default}}^T = \{x_c^T\}_{c=1}^C$ as the semantic prompts and a grid of points $X_{\text{default}}^S = \{x_n^S\}_{n=1}^N$ as the spatial prompts), SFMs (Zou et al., 2023) can predict a set of segmentation masks for $x^I$:

$$(M, C) = Decoder(z^I | Z_{\text{default}}^S, Z_{\text{default}}^T), \tag{2}$$

where $Z_{\text{default}}^S = Encoder^S(X_{\text{default}}^S)$ and $Z_{\text{default}}^T = Encoder^T(X_{\text{default}}^T)$.

On the other hand, directly using default prompts for every downstream dataset is usually sup-optimal, and how to acquire suitable prompts for SFMs is a non-trivial task but largely under-explored. In this work, we focus on investigating how to learn effective prompts for SFMs using few-shot data.

### 3.2 A Revisit of Prompt Learning

Prompt Learning aims to adapt a foundation model towards downstream tasks by optimizing the prompts using few-shot data. In recent years, various prompt learning methods have been proposed for image classification foundation models (CFMs) (Radford et al., 2021; Parisot et al., 2023). The core idea of CFM prompt leaning methods is to learn effective context text tokens to append and improve the raw class names, for better prompting the text encoder. Specifically, CFM prompt leaning methods, such as CoOp, introduce $M$ learnable context text tokens, i.e., $x_{\text{context}}^T = \{x_1^T, x_2^T, ..., x_M^T\}$, to model the context of each raw class name $x^T \in X_{\text{default}}^T$, such that the text prompts become $X_{CoOp}^T = \{X_{\text{default}}^T, X_{\text{context}}^T\}$. Given an image $x^I$, the text prompt $X_{CoOp}^T$ and a CFM consisting of an image encoder $Encoder^I$ and a text encoder $Encoder^T$, the image classification prediction can be formulated by:

$$c = Encoder^I(x^I) \cdot Encoder^T(\{X_{\text{default}}^T, X_{\text{context}}^T\}), \tag{3}$$

where '$\cdot$' denotes the inner (dot) product that measures the similarity between the image embedding and text embeddings. $X_{\text{default}}^T$ and $X_{\text{context}}^T$ are concatenated categorically before being fed into the text encoder.

To adapt CFMs to a downstream dataset, an image classification loss can be employed as the learning objective to optimize $X_{\text{context}}^T$ over few-shot data while keeping all other modules unchanged.

Different from previous works that focus on image classification foundation models and optimize text tokens to prompt text encoder, we examine the architecture of segmentation foundation models (SFMs) and propose a more efficient and effective prompt learning method for SFMs. Specifically, we optimize spatial and semantic prompts in the embedding space and selectively exploit the knowledge encoded in prompt encoder, ultimately learning effective prompt embeddings to prompt the mask decoder.

### 3.3 SPATIAL-SEMANTIC PROMPT LEARNING

We focus on prompt learning for SFMs using few-shot data. By examining the architecture of SFMs, we identify two challenges of learning effective prompts for SFMs: (1) *Limited Search Space in Spatial Prompt Learning.* (2) *Side Effects from Text Prompt Encoder.* We propose to tackle the two challenges by 1) directly optimizing prompts on the embedding space and 2) selectively leveraging the knowledge encoded in the pretrained prompt encoder. To this end, we design spatial-semantic prompt learning (SSPrompt) that introduces spatial prompt learning (SpaPrompt) and semantic prompt learning (SemPrompt), as illustrated in Figure 2. The two prompt learning methods complement each other by capturing spatial and semantic information respectively, which together learn effective spatial and semantic prompts for SFMs.

**Spatial prompt learning** (SpaPrompt) optimizes spatial prompts directly on the embedding space and selectively leverages the knowledge encoded in the pretrained spatial prompt encoder: it employs learnable weights to weight the default spatial prompt embeddings (encoded by the fixed spatial prompt encoder) and fuses the weighted embeddings with a set of learnable spatial prompt embeddings to acquire new spatial prompts. In this way, SpaPrompt learns effective spatial prompts for SFMs with two desirable features: 1) It relaxes the limited search space by optimizing spatial prompts directly on high-dimensional embedding space that has larger search space (e.g., 512 dimensions) than 2D coordinate space (i.e., 2 dimensions); 2) Similar to SemPrompt (mentioned in latter paragraphs), SpaPrompt can selectively utilize the knowledge encoded in the pretrained spatial prompt encoder.

Let $\hat{Z}^S = \{\hat{z}_n^S\}_{n=1}^N$ denote $N$ learnable spatial embeddings, where $\hat{z}_n^S \in \mathbb{R}^D$ and $D$ denotes the demension of embedding, and $\hat{W}^S = \{\hat{w}_n^S\}_{n=1}^N$ denote $N$ learnable weights, where $\hat{w}_n^S \in [0, 1]$. The new spatial prompt $Z_{\text{SpaPrompt}}^S$ can be obtained by applying $\hat{W}^S$ to weight the default spatial prompt embeddings $Z_{\text{default}}^S$ and fusing the weighted embeddings with $\hat{Z}^S$:

$$Z_{\text{SpaPrompt}}^S = \{\hat{w}_n^S \hat{z}_n^S + (1 - \hat{w}_n^S)z_n^S\}_{n=1}^N, \ \hat{z}_n^S \in \hat{Z}^S, \ z_n^S \in Z_{\text{default}}^S \text{ and } \hat{w}_n^S \in \hat{W}^S. \quad (4)$$

Given an image $x^I$ and the new spatial prompt $Z_{\text{SpaPrompt}}^S$, SFMs predict a set of segmentation masks:

$$(M, C) = Decoder(z^I | Z_{\text{SpaPrompt}}^S, Z_{\text{default}}^T), \quad (5)$$

where we can employ a segmentation loss to optimize $Z_{\text{SpaPrompt}}^S$ to find the best spatial prompts for SFMs with respect to each downstream dataset. Note, during training, we only update the learnable embeddings $\hat{Z}^S$ and the learnable weights $\hat{W}^S$ to optimize $Z_{\text{SpaPrompt}}^S$, while all other modules have been frozen as illustrated in Figure 2.

**Semantic prompt learning** (SemPrompt) optimizes semantic prompts directly on the embedding space and selectively leverages the knowledge encoded in the pretrained text prompt encoder: it employs learnable weights to weight the default semantic prompt embeddings (encoded by the fixed text prompt encoder) and fuses the weighted embeddings with a set of learnable semantic prompt embeddings to acquire new semantic prompts. SemPrompt learns semantic prompts for SFMs with two desirable features: 1) It is efficient as its optimization only involves the embeddings encoded by the large text prompt encoder instead of the text prompt encoder itself; 2) Its design of learnable weights allows to selectively leverage the semantic knowledge in the default semantic prompt embeddings and the learnable semantic prompt embeddings, which helps capture complementary knowledge, i.e., the former is encoded by the fixed text prompt encoder (pre-trained on large-scale data) and captures general semantic knowledge, while the latter is optimized and learnt from the downstream data and largely captures task-specific and domain-specific semantic knowledge.

Let $\hat{Z}^T = \{\hat{z}_c^T\}_{c=1}^C$ denote $C$ learnable semantic embeddings, where $\hat{z}_c^T \in \mathbb{R}^D$ and $D$ denotes the dimension of embedding, and $\hat{W}^T = \{\hat{w}_c^T\}_{c=1}^C$ denote $C$ learnable weights, where $\hat{w}_c^T \in [0, 1]$. The new semantic prompt $Z_{\text{SemPrompt}}^T$ can be obtained by applying $\hat{W}^T$ to weight the default semantic prompt embeddings $Z_{\text{default}}^T$ and fusing the weighted embeddings with $\hat{Z}^T$:

$$Z_{\text{SemPrompt}}^T = \{\hat{w}_c^T \hat{z}_c^T + (1 - \hat{w}_c^T) z_c^T\}_{c=1}^C, \ \hat{z}_c^T \in \hat{Z}^T, \ z_c^T \in Z_{\text{default}}^T \text{ and } \hat{w}_c^T \in \hat{W}^T. \tag{6}$$

Given an image $x^I$ and new semantic prompt $Z_{\text{SemPrompt}}^T$, SFMs predict a set of segmentation masks:

$$(M, C) = Decoder(z^I | Z_{\text{default}}^S, Z_{\text{SemPrompt}}^T), \tag{7}$$

where we can employ a segmentation loss to optimize $Z_{\text{SemPrompt}}^T$ to find the best semantic prompts for SFMs with respect to each downstream dataset. Note, during training, we only update the learnable embeddings $\hat{Z}^T$ and the learnable weights $\hat{W}^T$ to optimize $Z_{\text{SemPrompt}}^T$, while all other modules have been frozen as illustrated in Figure 2.

**Spatial-semantic prompt learning** (SSPrompt) combines spatial prompt learning and semantic prompt learning, aiming for leveraging the synergy of spatial and semantic information to better prompt segmentation foundation models. Given an image $x^I \in X^I$ and its segmentation annotation $y^I \in Y^I$, the new spatial prompt $Z_{\text{SpaPrompt}}^S$ from Eq. 4 and the new semantic prompt $Z_{\text{SemPrompt}}^T$ from Eq. 6, SSPrompt can be formulated as:

$$\{M, C\} = Decoder(z^I | Z_{\text{SpaPrompt}}^S, Z_{\text{SemPrompt}}^T), \tag{8}$$

$$\underset{\{\hat{Z}^S, \hat{W}^S, \hat{Z}^T, \hat{W}^T\}}{\arg\min} \frac{1}{|X^I|} \sum_{x^I \in X^I} \mathcal{L}_{seg}(\{M, C\}, y^I), \tag{9}$$

where $\mathcal{L}_{seg}$ denotes a standard segmentation loss function and $z^I = Encoder^I(x^I)$. Note we initialize $Z_{\text{default}}^S$ and $Z_{\text{default}}^T$ as in Eq. 2 before training such that the training process of SSPrompt will not involve the spatial prompt encoder and the large text prompt encoder.

## 4 EXPERIMENTS

Table 1: Datasets used to benchmark prompt learning for segment foundation models.

| Dataset | Classes | Images | Description |
|---|---|---|---|
| Cityscapes (Cordts et al., 2016) | 19 | 5,000 | Street scene images ($\sim$1080p) from European cities under good weather conditions. |
| BDD100K (Yu et al., 2020) | 19 | 10,000 | Street scene images ($\sim$720p) from American cities under various weather conditions. |
| Mapillary (Neuhold et al., 2017) | 19 | 25,000 | Street scene images from all over the world with high resolutions, e.g., $4000 \times 5000$ |
| ADE20K (Zhou et al., 2017) | 150 | 27,574 | A large-scale dataset with 20K+ scene-centric images and 150 semantic categories. |
| Pascal Context (Mottaghi et al., 2014) | 59 | 10,103 | An extension of the PASCAL VOC 2010 detection challenge with pixel-wise labels. |
| ACDC (Sakaridis et al., 2021) | 19 | 4,006 | An adverse conditions dataset with fog, nighttime, rain, and snow conditions. |

### 4.1 DATASETS

We benchmark our SSPrompt extensively over 6 widely used image segmentation datasets with pixel-wise annotations. As Table 1 shows, the 6 datasets have rich diversity, spanning from street scene data that include high-resolution images captured in different cities and under various daytimes, weathers and seasons, to category-rich data that cover 59 and 150 semantic categories. We did not include COCO dataset in experiments as it has been used in SFMs pre-training Zou et al. (2023).

### 4.2 IMPLEMENTATION DETAILS

We conduct experiments with two vision backbones including Focal-Tiny (Yang et al., 2022) and Davit-Large (Ding et al., 2022). In training, we employ SGD optimizer Loshchilov & Hutter (2017) with a weight decay of $1e - 4$, and set the base learning rate as $1e - 3$ which is adjusted with a polynomial learning rate schedule with a power of $0.9$. We use 1 GPU with batch size 2 for Cityscapes, BBD and ACDC, and 4 GPUs with batch size 8 for large datasets Mapillary, ADC20K and PASCAL Context. Our prompt learning method introduces very limited computation overhead, as illustrated

in Table 7 and appendix. We set the shorter side of input images at $512$ and employ random flip as data augmentation. The number of semantic prompts $C$ is set as the number of categories of each downstream dataset. Following Zou et al. (2023), we set the number of spatial prompts $N$ as 100 and the dimension of embedding $D$ as $512$. Following Chen et al. (2017), we employ cross-entropy loss as semantic segmentation loss. For instance segmentation and panoptic segmentation, we use multi-category cross-entropy loss for class prediction training and binary cross-entropy loss for mask prediction training.

Table 2: Prompt learning of segmentation foundation models on common datasets. The experiments are conducted on semantic segmentation (in mIoU), where 16-shot data are used (i.e., 16 labelled images for each class) for each dataset.

| Experiments with Tiny Vision Backbone | | | | | |
|---|---|---|---|---|---|
| Method | Cityscapes | BDD100K | Mapillary | ADE20K | PASCAL Context |
| SEEM-T (Zou et al., 2023) | 39.2 | 37.4 | 42.1 | 14.6 | 45.1 |
| CoOp (Zhou et al., 2022b) | 50.1 | 41.6 | 43.3 | 17.6 | 45.9 |
| LOCN (Parisot et al., 2023) | 51.5 | 42.6 | 44.2 | 19.3 | 46.6 |
| SSPrompt (Ours) | 55.2 | 47.1 | 49.5 | 23.2 | 51.2 |
| **Experiments with Large Vision Backbone** | | | | | |
| Method | Cityscapes | BDD100K | Mapillary | ADE20K | PASCAL Context |
| SEEM-L (Zou et al., 2023) | 49.3 | 44.6 | 47.9 | 15.2 | 37.1 |
| CoOp (Zhou et al., 2022b) | 51.2 | 45.2 | 52.0 | 18.1 | 47.4 |
| LOCN (Parisot et al., 2023) | 52.7 | 45.7 | 53.2 | 19.7 | 48.9 |
| SSPrompt (Ours) | 57.1 | 49.5 | 56.2 | 25.6 | 55.3 |

### 4.3 PROMPT LEARNING FOR SFMS ON COMMON DATASETS

Table 2 reports the image segmentation results on 5 widely-used common datasets. It can be seen that our SSPrompt achieves superior prompt learning performance consistently over various segmentation datasets. The superior performance is largely attributed to our two prompt learning designs that effectively address the two identified challenges in prompt learning for SFMs. Besides, it is expected that the large model SEEM-L should outperform the small model SEEM-T while SEEM-L performs unexpectedly not well on PASCAL Context dataset, where all prompt learning methods improve the performance while our SSPrompt brings the most substantial performance gain, showing that SSPrompt can well handle the occasional failures of SFMs.

Table 3: Prompt learning of segmentation foundation models on adverse-condition dataset, i.e., ACDC (Sakaridis et al., 2021). The experiments are conducted on semantic segmentation (in mIoU), where 16-shot data are used (i.e., 16 labelled images for each class) for each condition.

| Experiments with Tiny Vision Backbone | | | | | |
|---|---|---|---|---|---|
| Method | Foggy Condition | Night Condition | Rain Condition | Snow Condition | Mean |
| SEEM-T (Zou et al., 2023) | 34.6 | 26.2 | 33.1 | 35.8 | 32.4 |
| CoOp (Zhou et al., 2022b) | 36.7 | 28.6 | 33.5 | 36.4 | 33.8 |
| LOCN (Parisot et al., 2023) | 40.1 | 29.1 | 34.1 | 36.6 | 35.0 |
| SSPrompt (Ours) | 47.5 | 32.1 | 39.9 | 43.1 | 40.6 |
| **Experiments with Large Vision Backbone** | | | | | |
| Method | Foggy Condition | Night Condition | Rain Condition | Snow Condition | Mean |
| SEEM-L (Zou et al., 2023) | 48.1 | 32.0 | 47.4 | 45.0 | 43.1 |
| CoOp (Zhou et al., 2022b) | 52.2 | 33.5 | 48.2 | 45.6 | 44.9 |
| LOCN (Parisot et al., 2023) | 53.7 | 33.8 | 49.5 | 45.9 | 45.7 |
| SSPrompt (Ours) | 57.7 | 37.8 | 54.5 | 49.5 | 49.9 |

### 4.4 PROMPT LEARNING FOR SFMs ON ADVERSE-CONDITION DATASET

Table 3 reports the image segmentation results over the adverse-condition dataset, i.e., ACDC (Sakaridis et al., 2021). We can observe that SSPrompt outperforms the state-of-the-art by large margins consistently over different adverse conditions, demonstrating its great potential for more robust SFMs by learning effective domain-specific prompts.

### 4.5 DISCUSSION

**Generalization across different datasets.** We examine the generalization of SSPrompt with respect to image segmentation datasets. Specifically, we perform extensive evaluations over 6 widely studied common and adverse-condition datasets as described in Table 1. Experimental results in Tables 2- 3 show that SSPrompt achieves superior performance consistently across different types of image data.

**Generalization across different vision backbones.** We study the generalization of SSPrompt by evaluating it with two vision backbones, including Focal-Tiny (Yang et al., 2022) and Davit-Large (Ding et al., 2022). Results in Tables 2- 3 show that SSPrompt works effectively and consistently over both small and large vision backbones. Note we did not conduct experiments using SAM (Kirillov et al., 2023) as its version with text prompt encoder is not open-sourced and the released SAM version can only support class-agnostic segmentation.

**Generalization across different tasks.** We also examine the generalization of SSPrompt over different segmentation tasks including semantic segmentation, instance segmentation and panoptic segmentation. As Table 4 shows, SSPrompt improves the performance across all three segmentation tasks consistently. All experiments are conducted under the same setup with 16-shot data.

Table 4: Results on semantic (mIoU), instance (AP50) and panoptic segmentation (PQ).

| Cityscapes | Sem. Seg | Ins. Seg | Pan. Seg |
|---|---|---|---|
| SEEM-T | 39.2 | 32.7 | 32.4 |
| SSPrompt | 55.2 | 37.7 | 38.0 |

Table 5: Ablation studies of SSPrompt on Cityscapes dataset using 16-shot data.

| Method | Spatial Prompt Learning | | Semantic Prompt Learning | | mIoU |
|---|---|---|---|---|---|
| | Learnable Prompt Embedding | Learnable Prompt Weight | Learnable Prompt Embedding | Learnable Prompt Weight | |
| SEEM-T | | | | | 39.2 |
| | ✓ | | | | 46.2 |
| | ✓ | ✓ | | | 49.3 |
| | | | ✓ | | 51.5 |
| | | | ✓ | ✓ | 53.1 |
| SSPrompt | ✓ | ✓ | ✓ | ✓ | 55.2 |

**Ablation study.** We conduct ablation studies with Focal-Tiny on Cityscapes as shown in Table 5. We examine how SSPrompt's two core designs, i.e., 1) directly optimizing prompts on embedding space and 2) selectively leveraging the knowledge in prompt encoders, contribute to the overall performance. As Table 5 show, directly optimizing prompts on embedding space (i.e., optimizing learnable prompt embedding and average it with the default prompt embedding) improves the performance clearly, demonstrating its effectiveness on both spatial prompt learning and semantic prompt learning for better image segmentation with SMFs. In addition, instead of simply averaging, introducing learnable weights to selectively weight and fuse the default prompt embedding and the learnable prompt embedding brings further performance improvements, indicating that the learnable weights enable more effective usage of prompt encoders' knowledge and help learn better prompts. Moreover, combining spatial and semantic prompt learning performs the best clearly, demonstrating that the two types of prompt learning methods complement each other by providing orthogonal spatial and semantic information.

**Performance versus the number of training data.** We investigate how the amount of training data affects the performance by reducing it from 16-shot to 4-shot progressively. As shown in Table 6, SSPrompt still brings clear performance improvements against the baseline SEEM-T with less training data, showing the effectiveness of SSPrompt on different amounts of training data.

Table 6: Performance (in mIoU) versus number of data. The default is marked in  gray .

| SEEM-T | SSPrompt | | | |
|---|---|---|---|---|
| | 16-shot | 12-shot | 8-shot | 4-shot |
| 39.2 | 55.2 | 52.6 | 50.6 | 50.1 |

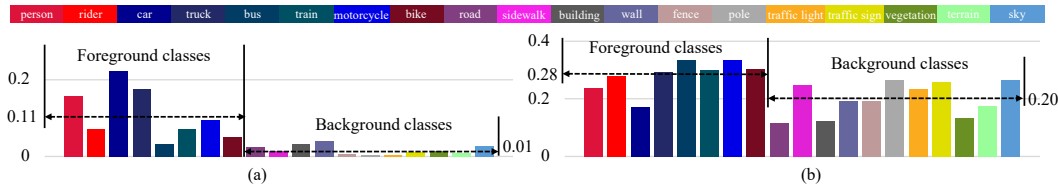

Figure 3: (a) Text data statistics (used for text prompt encoder pre-training in SFMs (Zou et al., 2023; Kirillov et al., 2023)). (b) Learnt weights in semantic prompt learning.

**Training efficiency comparison.** We analyze training efficiency by comparing prompt learning methods in training time (millisecond per image) and training memory (GB). Table 7 shows the results on ADE20K, indicating that SSPrompt is more efficient in training time and training memory. The superior efficiency is largely because SSPrompt circumvents the large text prompt encoder and requires less computation and memory. More results on other datasets are provided in the appendix.

Table 7: Training efficiency comparison in time (ms per image) and memory (GB).

| SEEM-T | CoOp | LOCN | SSPrompt |
|---|---|---|---|
| Training Time | 87.5 | 89.5 | 56.0 (-36.0%) |
| Training Memory | 8.22 | 8.22 | 3.82 (-53.5%) |

**Side Effects from Text Prompt Encoder.** We investigate the bias of Text Prompt Encoder and its side effects by comparing the text data statistics used to pre-train it, i.e., the occurrence of each class names in widely-used image-text dataset, LAION (Schuhmann et al., 2021). As shown in Figure 3 (a), the foreground class names generally occur much more frequently than background class names, which indicates that the text knowledge learnt from these data (i.e., the knowledge encoded in text prompt encoder) would bias toward foreground objects, leading to well-learnt foreground text knowledge but relatively poorly-learnt background text knowledge. Consequently, learning semantic prompts with such text prompt encoders can benefit from the well-learnt text knowledge, but may also suffer from the side effects from the poorly-learnt text knowledge. This is aligned with the ablation studies in Table 5, where introducing learnable weights to selectively exploit prompt encoder's knowledge improves the segmentation performance clearly. In addition, Figure 3 (b) visualizes the learnt weights in semantic prompt learning, where background classes are generally assigned with lower weights while foreground classes are often assigned with higher weights, showing that the foreground knowledge in text prompt encoder is more helpful in semantic prompt learning while background knowledge is less helpful.

**Limited Search Space.** We investigate how much the Limited Search Space issue affects learning spatial prompts by implementing Vanilla Spatial Prompt Learning (VSPL) that optimizes spatial prompts in 2D coordinate system. Results in Table 8 show that VSPL does not help much, largely due to the limited search space in VSPL. On the other hand, our SpaPrompt (and SSPrompt) optimizes prompts directly on high-dimensional embedding space, leading to larger search space and clearly improved performance.

Table 8: Comparison with Vanilla Spatial Prompt Learning (VSPL) on 16-shot data in mIoU.

| Method | SEEM-T VSPL | SpaPrompt | SSPrompt |
|---|---|---|---|
| Cityscapes | 39.2  41.0 | 49.3 | 55.2 |

Due to the space limit, we provide more dataset details, experiments and discussions in the appendix.

## 5 CONCLUSION

In this work, we identify two challenges of learning effective prompts for SFMs by examining the architecture of SFMs, and propose SSPrompt that tackles the identified challenges to learn effective semantic and spatial prompts for SFMs. Specifically, SSPrompt introduces spatial prompt learning and semantic prompt learning, which optimize spatial prompts and semantic prompts directly over the embedding space and selectively leverage the knowledge encoded in pre-trained prompt encoders. The two prompt learning methods complement each other by capturing spatial and semantic information respectively, which together learn effective spatial and semantic prompts for SFMs. Extensive experiments show that SSPrompt achieves superb image segmentation performance consistently across multiple widely adopted datasets. Moving forward, we will further explore prompt learning for better prompting SFMs.

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

# A APPENDIX

## A.1 DATASET DETAILS

We benchmark our SSPrompt extensively over 6 widely used image segmentation datasets with pixel-wise annotations. As Table 1 shows, the 6 datasets have rich diversity, spanning from street scene data that include high-resolution images captured over different cities and under various daytimes, weathers and seasons, to category-rich data that cover 59 and 150 semantic categories.

**Cityscapes** (Cordts et al., 2016) is a dataset designed for visual recognition tasks focused on urban street scenes. This dataset includes a training subset with 2,975 samples and a evaluation subset with 500 samples. Each image in both subsets is annotated at the pixel level, with labels assigned to 19 categories.

**BDD100K** (Yu et al., 2020) is a comprehensive dataset tailored for autonomous driving and urban scene analysis. This dataset consists of 7,000 training images and 1,000 validation images collected from various weather conditions, times of day, and urban landscapes, all of which are with pixel-wise annotations of 19 categories.

**Mapillary** (Neuhold et al., 2017) is a dataset primarily designed for urban scene understanding. This dataset contains 25,000 high-resolution images (e.g., 4000 x 5000) collected from all over the world with pixel-wise annotations. Following prior transfer learning work (Huang et al., 2021), we report results over 19 categories shared with Cityscapes.

**ADE20K** (Zhou et al., 2017) is a large-scale dataset with 27,574 scene-centric images which consists of 150 categories. This dataset consists of 25,574 training images and 2,000 validation images with pixel-wise annotations.

**Pascal Context** (Mottaghi et al., 2014) is an extension of PASCAL VOC 2010 detection dataset Everingham et al. (2010), which contains 59 categories with pixel-wise annotations. It has 4,998 training images and 1,449 validation images.

**ACDC** (Sakaridis et al., 2021) is a dataset designed for robust visual perception. ACDC consists of a large set of 4006 images collected from four common adverse conditions, i.e., fog, nighttime, rain, and snow. For each adverse condition, images are provided with high-quality pixel-level annotation of 19 categories.

## A.2 MORE DISCUSSION

Table 9: Training efficiency comparison in time (millisecond per image) and memory (GB).

| Dataset | Metric | CoOp | LOCN | SSPrompt |
|---|---|---|---|---|
| Mapillary (Cityscape, BDD100K, ACDC) | Training Time (ms/img) | 155.0 | 158.0 | 122.4 (-21.0%) |
| | Training Memory (GB) | 4.52 | 4.52 | 3.22 (-28.7%) |
| PASCAL Context | Training Time (ms/img) | 122.1 | 122.4 | 83.4 (-31.6%) |
| | Training Memory (GB) | 4.30 | 4.30 | 2.49 (-42.0%) |
| ADE20K | Training Time (ms/img) | 87.5 | 89.5 | 56.0 (-36.0%) |
| | Training Memory (GB) | 8.22 | 8.22 | 3.82 (-53.5%) |

**Full results of training efficiency comparison.** We analyze training efficiency by comparing prompt learning methods in training time (millisecond per image) and training memory (GB). Results in Table 9 (over SEEM-T model) show that our SSPrompt is more efficient in training time and training memory, largely because the optimization of SSPrompt circumvents the large text prompt encoder and requires less computation and memory, i.e., it only involves the embeddings encoded by the large text prompt encoder instead of the text prompt encoder itself. All results are measured under the same setup and device. The training time and training memory over various datasets are different as they include different numbers of categories and different image resolutions. Cityscapes, BDD100K, ACDC and Mapillary datasets have the same number of categories and their images will be resized into the same image resolution during training. Therefore, they share similar training time and memory, and we only report the ones for Mapillary for simplicity.

**Comparison with prompt engineering.** One common traditional method to tailor prompts for downstream datasets is by prompt engineering (Radford et al., 2021). We investigate how much prompt engineering affects the segmentation performance by comparing prompt learning methods with the baseline SEEM-T with and without prompt engineering. Note, in this paper, all the results of SEEM-T (and SEEM-L) are already with prompt engineering (i.e., averaging of 80 hand-crafted prompt templates (Zou et al., 2023)) unless specifically mentioned otherwise. Here, we remove such prompt engineering for comparison and analysis. As Table 10 shows, prompt learning methods, e.g., CoOp (Zhou et al., 2022b), LOCN (Parisot et al., 2023) and our SSPrompt, performs much better than hand-crafted prompt engineering. Besides, our SSPrompt performs the best clearly, showing that SSPrompt learns more effective prompts for segmentation foundation models as compared with either previous prompt learning methods or prompt-engineering.

Table 10: Comparison with baseline SEEM-T with and without prompt engineering on Cityscapes under 16-shot data setup.

| Method | 5-task Mean |
|---|---|
| SEEM-T w/o Prompt Engineering | 37.7 |
| SEEM-T w/ Prompt Engineering | 39.2 |
| CoOp | 50.1 |
| LOCN | 51.5 |
| SSPrompt | 55.2 |

|  SEEM-T | CoOp | LOCN | SSPrompt (Ours) | Ground Truth |

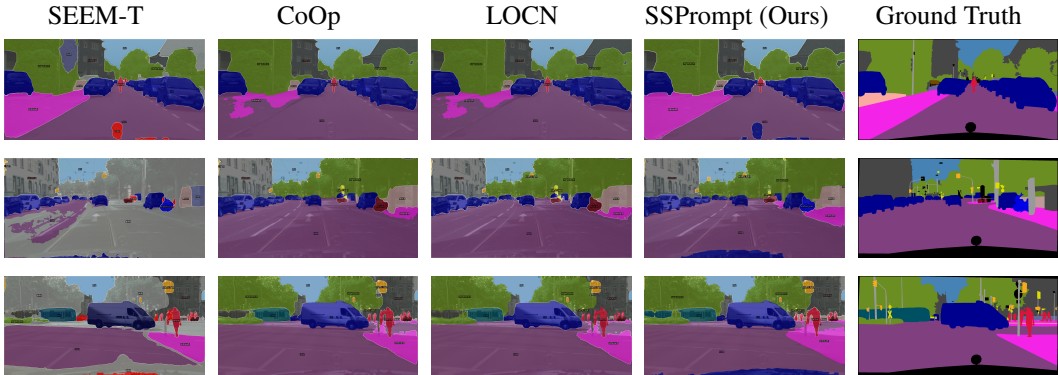

Figure 4: Qualitative illustration and comparison over prompt learning for segmentation foundation model (SFM). The experiments are conducted on semantic segmentation, where 16-shot data are used (i.e., 16 labelled images for each class) for each dataset. Our proposed spatial-semantic prompt learning (SSPrompt) tackles the identified two challenges by 1) directly optimizing prompts on the embedding space and 2) selectively leveraging the knowledge encoded in the pretrained prompt encoder, which learns effective semantic and spatial prompts that prompt SFMs to generate more accurate segmentation results. It can be observed that SSPrompt produces better segmentation results, for example, the sidewalk predictions from SSPrompt are less noisy and have better outlines.

**Qualitative results.** We provide qualitative results in Figure 4. Our proposed spatial-semantic prompt learning (SSPrompt) tackles the identified two challenges by 1) directly optimizing prompts on the embedding space and 2) selectively leveraging the knowledge encoded in the pretrained prompt encoder, which learns effective semantic and spatial prompts that prompt SFMs to generate more accurate segmentation results. It can be observed that SSPrompt produces better segmentation results, for example, the sidewalk predictions from SSPrompt are less noisy and have better outlines.

## A.3 RELATIONS TO VISUAL PROMPT LEARNING

In this work, we focus on learning effective prompts for segmentation foundation models (SFMs) using few-shot data, aiming to facilitate the deployment of SFMs for task-specific or domain-specific downstream datasets. As there is little previous work in this field, we compare our proposed SSPrompt with text prompt learning methods of classification foundation model (Zhou et al., 2022b; Parisot et al., 2023) for benchmarks.

We note that visual prompt tuning (Jia et al., 2022; Zang et al., 2022) also focuses on improving classification foundation models (CFMs) using few-shot data, which learns additional image pixels/patches to append and modify the raw images (Jia et al., 2022; Zang et al., 2022), such that it

can mitigates the domain gaps in image distributions between CFM pre-training data and the target dataset.

Like text prompt learning methods (Zhou et al., 2022b; Parisot et al., 2023), our SSPrompt is orthogonal to visual prompt tuning methods (Jia et al., 2022; Zang et al., 2022), because SSPrompt and text prompt learning methods learn to modify prompts only (i.e., semantic prompts or spatial prompts) and do not modify any image content. On the contrary, visual prompt tuning methods focus on modifying image content properly to improve the model performance. Thus, we did not compare our SSPrompt with visual prompt tuning methods becuase these two types of methods work very differently and complement each other (Zang et al., 2022).

Note the spatial prompt learning in SSPrompt is quite different to visual prompt tuning: the former learns to provide effective location information to prompt SFMs for better performance, while the later learns to modify image content properly to improve the model performance.

### A.4    OTHER FINE-TUNING METHODS OF SFMS

In this work, we focus on learning effective prompts for segmentation foundation models (SFMs) using few-shot data, aiming to facilitate the deployment of SFMs for task-specific or domain-specific downstream datasets. This learning paradigm has several desirable features: 1). It is data efficient as it needs only few-shot data, e.g., 16-shot, 8-shot or 4-shot data. 2) It is computation efficient as it only learns to modify the prompts instead of the whole model. 3) It is generalizable due its simplicity, e.g., it works well on various segmentation tasks like semantic segmentation, instance segmentation and panoptic segmentation.

We note that there are several concurrent works that also focus on fine-tuning SFMs (Zhang et al., 2023; Liu et al., 2023b). In this section, we briefly introduce their fine-tuning setups and methods and clarify the difference between them and the studied setup.

For example, (Zhang et al., 2023) proposes to personalize SFM to automatically segment unique visual concepts. Specifically, (Zhang et al., 2023) design PerSAM, which takes an object image and its segmentation mask as the reference to customize SFM to segment this unique object in other images. Differently, we focus on general-purpose segmentation tasks like semantic segmentation, instance segmentation and panoptic segmentation, instead of the customized segmentation in (Zhang et al., 2023) which aims to segment the object with unique identity (i.e., the pet dog of a certain user).

Besides, (Liu et al., 2023b) proposes to improve SFM by combining it with an all-purpose feature extraction model (Oquab et al., 2023). Specifically, given a reference image with pixel-level annotations, the all-purpose feature extraction model (Oquab et al., 2023) could provide effective dense correspondences between the labelled reference image and a given test image, which are utilized by (Liu et al., 2023b) as the prompt to prompt SFM. Differently, our work directly optimizes the prompts with few-shot data and does not introduce any other large models, which is much more computation efficient as compared with (Liu et al., 2023b) that introduces an additional large model for better segmentation performance.

