# OpenReview forum: "Learning to Prompt Segmentation Foundation Models"
_ICLR.cc/2024/Conference — ICLR 2024 Conference Withdrawn Submission_

### Official Review · Reviewer_8VMA · 2023-10-23

**Soundness:** 2 fair
**Presentation:** 4 excellent
**Contribution:** 2 fair
**Rating:** 3
**Confidence:** 5

**Summary:**

This paper focuses on the topic of segmentation foundation models (SFMs) and their potential in learning to segment various objects. SFMs, such as SEEM and SAM, are built upon the concept of "Promptable Segmentation," which involves using a handcrafted prompt as input to obtain the desired segmentation mask.

SFMs utilize two types of prompts: spatial prompts (e.g., points) and semantic prompts (e.g., texts). These prompts work together to guide SFMs in segmenting objects in downstream datasets. However, the process of acquiring suitable prompts for SFMs has not been extensively explored.

To address this gap, the authors of this paper analyze the architecture of SFMs and identify two challenges related to prompt learning. In response, they propose a method called spatial-semantic prompt learning (SSPrompt) that aims to learn effective semantic and spatial prompts for improved SFMs.

SSPrompt introduces spatial prompt learning and semantic prompt learning techniques, which optimize spatial and semantic prompts directly in the embedding space. Additionally, the proposed approach selectively leverages the knowledge encoded in pre-trained prompt encoders.

The paper presents extensive experiments that demonstrate the superior performance of SSPrompt in image segmentation across multiple widely adopted datasets.

**Strengths:**

+ The paper introduces an approach called spatial-semantic prompt learning (SSPrompt) to address the challenge of acquiring suitable prompts for segmentation foundation models (SFMs). By optimizing prompts in the embedding space and leveraging pre-trained prompt encoders, SSPrompt offers a fresh perspective on improving SFMs' performance.

+ The paper exhibits a high level of quality in terms of its methodology and experimental evaluation. It delves into the challenges associated with prompt learning for SFMs and proposes specific techniques to tackle them. The proposed SSPrompt method is extensively evaluated on multiple widely adopted datasets, showcasing consistent superior performance in image segmentation. The experiments are well-designed, and the results are thoroughly analyzed and discussed.

**Weaknesses:**

- The paper has certain limitations in terms of its technical contribution. The introduction of learnable text prompt embeddings and spatial prompt embeddings to the SFMs baselines is relatively straightforward, as is the weighting strategy employed.

- Furthermore, the experimental evaluation falls short in providing comprehensive results. While the paper presents results on the SEEM model, it lacks experimentation and analysis on the SAM model. This omission restricts a comprehensive understanding of the proposed approach's effectiveness across different SFM architectures.

**Questions:**

Please refer to paper Weaknesses.

---

### Official Review · Reviewer_VfrA · 2023-10-31

**Soundness:** 3 good
**Presentation:** 3 good
**Contribution:** 2 fair
**Rating:** 6
**Confidence:** 4

**Summary:**

This paper proposes a SSPormpt method to adapt large-scale pre-trained segmentation fundation models for different segmentation tasks. The model shows strong performance across a range of different datasets.

**Strengths:**

1. The paper is well written, and the motivation and the method are clearly explained.
2. The proposed method makes interesting use of feature spces to adapt default prompts for SFMs. Although there exists several works done in other research fields, this seems novel in SFMs.
3. The experiments are impressively comprehensive, covering a wide range of different datasets.
4. The method shows very strong performance across all datasets.
5. The authors also consider a number of different ablations to better understand the proposed method.

**Weaknesses:**

1. I think is a new method, but is not novel enough. From my point of view, it is an incremental method with good performance.

2. Though the ablation experiments are extensive, it seems that there may be a very important ablation that was not performed. From my experiences, the number of prompts is very important for prompt learning. However, the authors do not show the results obtained by different number of prompts. I think it is important to perform a targeted ablation about that.

3. I am also curious to see what the meaning of each learned prompt. The authors are suggested to present some analysis about the learned prompt like CoOp.

4. There is no failure cases, which I believe is very important for reader to understand the proposed technique.

**Questions:**

Additional:

The authors are suggested to do cross-dataset generalization to demonstrate the goodness of the proposed method.

---

### Official Review · Reviewer_3WRG · 2023-11-05

**Soundness:** 3 good
**Presentation:** 3 good
**Contribution:** 4 excellent
**Rating:** 5
**Confidence:** 4

**Summary:**

In this paper, authors focus on segmentation foundation models (SFM), and identify two challenges of learning effective prompts. They propose SSPrompt to learn effective semantic and spatial prompts for better SFMs, introducing spatial prompt learning and semantic prompt learning strategies. And the proposed method shows compelling performance.

**Strengths:**

1. The problem of prompting learning for SFM is important. This paper takes a good step in this direction. The overall framework makes sense to me.
2. The paper is well written and easy to follow.
3. The results are strong across all segmentation datasets.
4. The contributions are well ablated.

**Weaknesses:**

1. The paper seems to possess marginal novelty. The weighting strategy used in learnable prompt embeddings is quite straightforward.
2. The ablations of applying your strategy on other SFMs such as SAM are missing. This may affect the generality of your proposed strategy.
3. Performance regarding the mIoU of foreground and background classes should be displayed in addition to the learned weights. The former is more straightforward to illustrate your contribution.

**Questions:**

Additional:
Fig.1 and Fig.2 could combine rather than splitting into two, which is a waste of space, and the Table. 2 should combine as well.
Cross-dataset generalization is also suggested.

---

### Official Review · Reviewer_V4Gx · 2023-11-07

**Soundness:** 3 good
**Presentation:** 2 fair
**Contribution:** 3 good
**Rating:** 6
**Confidence:** 4

**Summary:**

The paper introduces a novel approach called spatial-semantic prompt learning (SSPrompt) for enhancing the performance of Segmentation Foundation Models (SFM). By identifying a previously under-explored aspect of SFM - the acquisition of effective prompts - the authors propose an optimization method for both spatial and semantic prompts in the embedding space. The method distinguishes itself by not requiring additional large models, thus improving computational efficiency. The authors claim that SSPrompt consistently outperforms existing methods across various datasets and demonstrates strong generalization across different vision backbones. While the paper presents promising results, a full assessment of the experimental results and a comparison with state-of-the-art methods would be crucial for a complete evaluation. Overall, the paper suggests a potentially significant advancement in image segmentation, with implications for the deployment of SFMs in a wide range of applications.

**Strengths:**

The paper presents an innovative spatial-semantic prompt learning (SSPrompt) method for Segmentation Foundation Models, introducing a novel approach that creatively combines spatial and semantic prompts for better performance. The quality of the research appears robust, with a clear methodology and promising preliminary results that suggest significant improvements in computational efficiency and generalizability across datasets. While the paper is well-structured and articulates its contributions with clarity, further clarity could be achieved through detailed experimental results and comparisons with state-of-the-art methods. The significance of this work is potentially high, given its implications for the practical deployment of SFMs in a wide range of applications. Overall, if the empirical results are validated thoroughly, this paper could represent a substantial advancement in the field of image segmentation.

**Weaknesses:**

The paper's innovative approach to improving SFMs through SSPrompt is promising but could be enhanced by providing more comprehensive experimental results. A rigorous comparative analysis with state-of-the-art methods, complete with benchmarks and quantitative data, is essential to establish the method's relative standing. Further methodological justifications, such as the rationale behind dataset and backbone selection, would clarify the design choices. To ensure reproducibility and validate computational efficiency claims, the paper should include detailed implementation details, complexity analysis, and runtime comparisons.

**Questions:**

Could the authors elaborate on the implementation details of SSPrompt, including hyperparameters and training details?
What are the limitations of SSPrompt when faced with extremely varied or unseen data types?